# Epidemiological Characteristics and Antimicrobial Resistance Changes of Carbapenem-Resistant *Klebsiella pneumoniae* and *Acinetobacter baumannii* under the COVID-19 Outbreak: An Interrupted Time Series Analysis in a Large Teaching Hospital

**DOI:** 10.3390/antibiotics12030431

**Published:** 2023-02-22

**Authors:** Xinyi Yang, Xu Liu, Weibin Li, Lin Shi, Yingchao Zeng, Haohai Xia, Qixian Huang, Jia Li, Xiaojie Li, Bo Hu, Lianping Yang

**Affiliations:** 1School of Public Health, Sun Yat-sen University, Guangzhou 510080, China; 2Department of Infectious Disease, The Fifth Affiliated Hospital, Sun Yat-sen University, Zhuhai 519000, China; 3Department of Pharmacy, The First Affiliated Hospital, Sun Yat-sen University, Guangzhou 510080, China; 4Department of Laboratory Medicine, The Third Affiliated Hospital of Sun Yat-sen University, Guangzhou 510630, China

**Keywords:** antimicrobial resistance, multidrug-resistant organisms, nosocomial infection, COVID-19, interrupted time series

## Abstract

Background: To investigate the epidemiological characteristics and resistance changes of carbapenem-resistant organisms (CROs) under the COVID-19 outbreak to provide evidence for precise prevention and control measures against hospital-acquired infections during the pandemic. Methods: The distribution characteristics of CROs (i.e., carbapenem-resistant *Klebsiella pneumoniae* and *Acinetobacter baumannii*) were analyzed by collecting the results of the antibiotic susceptibility tests of diagnostic isolates from all patients. Using interrupted time series analysis, we applied Poisson and linear segmented regression models to evaluate the effects of COVID-19 on the numbers and drug resistance of CROs. We also conducted a stratified analysis using the Cochran–Mantel–Haenszel test. Results: The resistance rate of carbapenem-resistant *Acinetobacter baumannii* (CRAB) was 38.73% higher after the COVID-19 outbreak compared with before (*p* < 0.05). In addition, the long-term effect indicated that the prevalence of CRAB had a decreasing trend (*p* < 0.05). However, the overall resistance rate of *Klebsiella pneumoniae* did not significantly change after the COVID-19 outbreak. Stratified analysis revealed that the carbapenem-resistant *Klebsiella pneumoniae* (CRKP) rate increased in females (OR = 1.98, *p* < 0.05), those over 65 years old (OR = 1.49, *p* < 0.05), those with sputum samples (OR = 1.40, *p* < 0.05), and those in the neurology group (OR = 2.14, *p* < 0.05). Conclusion: The COVID-19 pandemic has affected the change in nosocomial infections and resistance rates in CROs, highlighting the need for hospitals to closely monitor CROs, especially in high-risk populations and clinical departments. It is possible that lower adherence to infection control in crowded wards and staffing shortages may have contributed to this trend during the COVID-19 pandemic, which warrants further research.

## 1. Introduction

Infections caused by multidrug-resistant organisms (MDROs) have been identified as a top global public health concern and a priority for hospital infection prevention and control. Among MDROs, carbapenem-resistant organisms (CROs), such as carbapenem-resistant Enterobacteriaceae (CRE; e.g., carbapenem-resistant *Klebsiella pneumoniae* (CRKP)) and glucose non-fermenting (NF) Gram-negatives (e.g., carbapenem-resistant *Acinetobacter baumannii* (CRAB)) are responsible for a significant proportion of nosocomial infections and pose a critical threat to inpatients due to limited treatment options and high mortality rates [1,2,3,4]. A Centers for Disease Control and Prevention (CDC) report identified multidrug-resistant Acinetobacter as a severe threat with death rates as high as 52% [5]. Another study reported a mortality rate of up to 47% for infections caused by CRE [6].

Despite significant investments and prevention efforts such as nosocomial infection and control bundles and antibiotic stewardship to protect against antimicrobial-resistant infections and their spread, a recent report from the US CDC indicated that significant increases in infections caused by antimicrobial-resistant organisms including CROs were observed during the COVID-19 pandemic [5]. The similarity of symptoms between COVID-19 and bacterial pneumonia, as well as the occurrence of secondary bacterial infections, may make it difficult for clinicians to adhere to antibiotic stewardship, potentially leading to irrational or increased antibiotic use in hospitalized patients and worsening antimicrobial resistance [7]. For example, a study from China found that 95% of COVID-19 patients in the hospital received antibiotics, even though only 15% had secondary bacterial infections [8]. A systematic review and meta-analysis of antimicrobial drug consumption also reported a significant increase in antibiotic use in primary and secondary health care in several high-income countries in the early stages of the COVID-19 pandemic [9].

There is increasing evidence that the effects of COVID-19 on MDROs vary, and specific antimicrobial stewardship measures should be prioritized over existing empirical treatment guidelines [10]. Understanding the epidemiological characteristics of CROs and the change in resistance is critical for informing effective measures to prevent and control infections and mitigate antimicrobial resistance (AMR) in response to COVID-19, and deserves a great deal of attention [11]. However, much of the current evidence is from high-income countries, and there is a lack of evidence from low- and middle-income countries (LMICs) where the healthcare systems are facing greater challenges from COVID-19 and CROs due to the lack of clinical microbiology laboratory capacity and deficiencies in infection prevention and control measures [3]. This study aimed to explore the distribution and resistance changes of CROs before and after the COVID-19 outbreak in order to strengthen public health and healthcare systems in LIMICs to respond to the threat of AMR.

## 2. Results

Table 1 shows the total proportion of common disease-causing bacteria in teaching hospitals. From April 2018 to September 2021, a total of 22,062 pathogenic strains of bacteria were isolated from patients at the hospital. Of these, 2495 strains of *K. pneumoniae* and 2240 strains of *A. baumannii* accounted for 11.31% and 10.15% of the total, respectively. The proportion of *A. baumannii* was significantly higher than the national data, while the proportion of *K. pneumoniae* was significantly lower than the national data, according to the China Antimicrobial Resistance Surveillance System (CARSS).

Appendix A presents the distribution of *A. baumannii*, CRAB, *K. pneumoniae*, and CRKP in different groups. At the population level, the proportions of *K. pneumoniae* and *A. baumannii* and the resistance rate of CRAB were higher in males than in females. The proportion of *A. baumannii* was the highest in those under 1 year old (29.78%). However, the resistance rate of CRAB was highest (71.61% to 78.86%) in patients 1 to 14 years old. The proportion of *K. pneumoniae* was also highest in patients under 1 year old, while the resistance rate of carbapenems was highest in those over 65 years old (21.52%). Sputum specimens were the primary source of *K. pneumoniae* (16.83%), with the highest resistance rate of CRKP in clean midstream urine (26.88%). The proportion of *A. baumannii* was highest in sputum specimens (23.33%), but the highest resistance rate of CRAB was in wound swabs (75.68%). At the department level, the resistance rate of *A. baumannii* was highest in the neonatology department (32.39%), followed by the ICU (27.34%) and neurology (21.07%). The proportion of *K. pneumoniae* was highest in neonatology (19.60%), followed by the infectious diseases department (19.45%) and sputum (16.83%). However, the resistance rates of CRAB and CRKP in the ICU (86.33% and 32.58%) were higher than in other departments. At the ward level, the resistance rate of CRKP in inpatients (20.03%) was higher than in outpatients, with no statistically significant difference in CRAB.

Figure 1 illustrates the temporal variation in the numbers and proportions of *A. baumannii* and *K. pneumoniae*. After January 2020, there was a significant decrease in the number of *K. pneumoniae* and *A. baumannii*. The temporal trends of CRKP and CRAB showed periodicity and seasonality. After the outbreak of COVID-19 in January 2020, there was a slight decrease in the resistance rates of CRKP and CRAB for a short period.

Table 2 and Figure 2 display the trends in the number and resistance rates of *A. baumannii*, CRAB, K. pneumonia, and CRKP, along with the results of the segmented regression analysis. The initial level of *A. baumannii* detected in the teaching hospital was 35.03 strains/2 weeks (*p* < 0.001). The level of *A. baumannii* before the COVID-19 outbreak showed a downward trend (*p* < 0.001) and the number of *A. baumannii* detections decreased by 1.56 strains/2 weeks after the outbreak compared with the baseline level of the observation period (*p* < 0.05), but the long-term effect showed an increasing trend (*p* < 0.01). The baseline resistance rate of CRAB was 77.12% (*p* < 0.001). The CRAB resistance rate increased by 38.73% after the COVID-19 outbreak compared with the baseline level of the observation period (*p* < 0.05), but the long-term effect showed a decreasing trend of 0.6%/2 weeks (*p* < 0.05).

The baseline level of *K. pneumoniae* detections was 30.34 strains/2 weeks (*p* < 0.001). After the COVID-19 outbreak, the number of *K. pneumoniae* detections decreased by 4.13 strains/2 weeks compared with the initial level (*p* < 0.001), but the long-term effect showed an increasing trend (*p* < 0.001). The baseline resistance rate of CRKP was 10.31% (*p* < 0.05). The resistance level of CRKP showed an increasing trend before the COVID-19 outbreak (*p* < 0.05), but the resistance rate and trend did not significantly change after the outbreak.

Appendix A shows that, after the removal of duplicate isolates per patient, the overall trend of detection number and resistance rate of *Acinetobacter baumannii* and *Klebsiella pneumoniae* during COVID-19 did not change. The results were consistent whether duplicate isolates per patient were removed or not for the resistance rate change of CRAB and CRKP. The time trends of pathogens and drug-resistant bacteria after removing duplicates per patient are shown in Appendix A.

Figure 3 illustrates the variation in the resistance rates of CRAB and CRKP among different subgroups before and after COVID-19. After the COVID-19 pandemic, the resistance rate of *A. baumannii* decreased significantly in females (OR = 0.48, *p* < 0.001), the 15- to 65-year-old group (OR = 0.68, *p* < 0.01), the > 65-year-old group (OR = 0.63, *p* < 0.01), the sputum sample group (OR = 0.70, *p* < 0.01), and the ICU group (OR = 0.34, *p* < 0.001). For CRKP, the results in the female group, > 65-year-old group, sputum sample group, and neurology group were statistically significant (*p* < 0.05), and the OR values were all greater than 1, indicating that the resistance rate of CRKP increased after the COVID-19 pandemic (OR = 1.98, OR = 1.49, OR = 1.40, and OR = 2.14, respectively).

## 3. Discussion

This study identified some high-risk groups and departments with high resistance pressure during the COVID-19 pandemic, providing evidence for precise prevention and control measures against hospital-acquired infections [3,12]. The age distribution showed a higher resistance rate of CRO in pediatric populations. Children with low body weight and incomplete immune systems are more susceptible to the invasion of external pathogens and nosocomial infection [13]. Additionally, limited types of antibiotics are available for children, so cephalosporins, carbapenems, and macrocyclic lipids are widely used, which may lead to drug resistance [14] The resistance rates of CRKP and CRAB were also found to be higher in the ICU compared with other departments, which is in line with previous research findings [2,15,16,17]. In developed countries, over 20% of nosocomial infections occur in the ICU [18], where patients are at a higher risk of MDRO infection due to low organ function; comprised immunity; and underlying diseases such as diabetes, chronic obstructive pulmonary disease, and chemotherapy [4]. Outbreaks of nosocomial infections in the ICU led to high mortality rates and should be closely monitored [19] with special attention.

The hospital, which serves as both a large tertiary referral center and a designated facility for treating local COVID-19 patients during the pandemic, has been affected by a number of factors that may have impacted the detection numbers of pathogens and the resistance rate of drug-resistant pathogens. The early phase of the pandemic saw changes in healthcare-seeking behavior and reduced access to healthcare, potentially leading to undiagnosed and untreated bacterial infections [20]. At this time, the resistance rate of CRAB increased, potentially due to the non-specific nature of COVID-19 and concerns about secondary bacterial infections, leading to increased irrational use of antibiotics and an increased risk of drug resistance and hospital-acquired infections. Additionally, the strain on medical resources and staffing shortages during the pandemic may have contributed to overcrowding in the hospital, increasing the spread of drug-resistant bacteria [21] and potentially leading to lower adherence to infection control measures. However, the long-term decline in the resistance rate of CRAB may be the result of the implementation of environmental cleaning and decolonization measures following the normalization of COVID-19, which can help to interrupt the transmission of CRAB and contribute to infection control [22]. As clinical treatment guidelines for COVID-19 are clarified and the use of antibiotics in COVID-19 treatment becomes more standardized, the risk of bacterial resistance may be reduced [23]. It is worth noting that this study used diagnostic isolates for statistical analysis, so the number of isolates was influenced by the number of hospitalized patients. Some research has suggested that hospital resources may have been redirected towards COVID-19 treatment, leading to decreased admissions and lower adherence to infection control measures, resulting in a lower specimen submission rate and slower detection and reporting of AMR data [5,24].

The COVID-19 pandemic did not significantly impact the resistance rate of CRKP. This is supported by a study in Italy which found no statistical difference in the incidence ratio, colonization rate, and infection rate of CRE during the pandemic [21]. However, other research has shown that, despite the implementation of stronger hospital infection control measures, the infection rate of CRKP in ICUs actually increased rather than decreased. The possible reason was that ICU patients often require intensive care and prolonged treatment [25], but further studies are needed to confirm the exact mechanism.

The differences in the changes of CRAB and CRKP before and after COVID-19 may be related to the distinct transmission mechanisms of these drug-resistant bacteria. *A. baumannii* is often colonized in healthcare settings, such as on shared equipment surfaces that are not properly cleaned, and can spread through contact with contaminated surfaces or hands. Enhanced hand hygiene and environmental disinfection during COVID-19 may have effectively reduced the transmission of *A. baumannii* [26], contributing to the decrease in the CRAB resistance rate seen in this study. On the other hand, *K. pneumoniae* is not transmitted through the air or through environmental contamination, so improvements in hospital environments during COVID-19 may not have had a significant effect. Additionally, many COVID-19 patients use devices such as ventilators or intravenous catheters, which can also increase the risk of *K. pneumoniae* infection [27].

According to the stratified analysis, the resistance rate of CRKP increased in the >65 year age group, sputum sample group, and neurology group after COVID-19. The CARSS report also shows that the resistance rate of CRKP has been increasing over the past five years [28], and a study at this teaching hospital found that the resistance rate of CRKP had a rising trend prior to COVID-19, which may continue after the pandemic. Older patients are generally more vulnerable to pathogens and many have underlying medical conditions that increase their risk of drug-resistant bacterial infections [29,30], making them more susceptible to the effects of a pandemic on infection control practices. Similarly, patients receiving neurologic care often require long-term hospitalization, surgical treatment, and intensive care due to conditions such as cerebral hemorrhage and cerebral infarction [31], which may also increase their risk of CRKP infection.

This study had several limitations that should be considered. First, the data were collected from a single tertiary hospital, so caution is needed when generalizing the findings to other settings. Second, the breakpoint for the study was chosen based on the WHO’s declaration of the pandemic, but the progress of the epidemic may vary in different regions. However, the response of the Chinese government and public hospitals to both international and domestic epidemic response has generally been timely. Third, the number of detected isolates could not be normalized due to the lack of data on the weekly number of patients during the study period. As a result, this study focused on comparing the resistance rates of CRAB and CRKP rather than the exact number of isolates. Finally, some detailed patient information such as antibiotic use, length of stay, invasive procedures, and complications were important but not available, which limited the ability to make further inferences.

## 4. Materials and Methods

### 4.1. Setting

This retrospective interrupted time series analysis was conducted at a tertiary teaching hospital in Guangdong Province, China, which has four wards with a total of 7400 beds and an average of 18,000 outpatient visits per day. The hospital has a range of departments and frequently receives referrals from other hospitals, making the collected microbiological data representative of a diverse group of patients and specimen types. The hospital’s clinical laboratory maintains electronic medical records of diagnostic isolates and antimicrobial susceptibility testing results, ensuring the integrity and accuracy of the data.

### 4.2. Data Collection

The data were collected from April 2018 to September 2021, covering the period before and after the COVID-19 outbreak. The collected data included information on all diagnostic isolates and drug susceptibility, as well as patient information such as patient ID, sex, age, ward type, department, source of the specimen, and species of infectious pathogen from the nosocomial infection surveillance system. Information on susceptibility to major antibiotics was obtained from the drug resistance monitoring network, and national data on CROs distribution and drug resistance were collected from CARSS for comparison. The CROs of focus in this study were CRAB and CRKP. Carbapenem resistant organisms were defined as organisms that are not susceptible to at least one class of carbapenem antibiotics, including imipenem, meropenem, and ertapenem. Isolate identification was conducted using VITEK MS bacterial mass spectrometry, and antimicrobial susceptibility testing was performed with the VITEK 2 Compact system. Test results were interpreted according to the association of Clinical and Laboratory Standard Institute (CLSI) criteria [32].

### 4.3. Statistical Analysis

The proportion of *K. pneumoniae* and *A. baumannii* (among all pathogen types) and their carbapenem resistance rates were calculated for the overall population and for subgroups based on different departments/wards, patient gender, age groups (i.e., <1 year, 1–14 years, 15–65 years, and >65 years according to the CARSS-V6.0 standard), and specimen sources (i.e., sputum, urine, blood, drain, and wound). These rates were compared using a Chi-square test and were also compared with national data from CARSS for the same year.

An interrupted time series (ITS) analysis was conducted to evaluate the impact of COVID-19 on the numbers and carbapenem resistance rates of *A. baumannii* and *K. pneumoniae*. We set two weeks as the time unit. Given the lag time between the emergence of resistance and the implementation of infection prevention and control measures by the hospital during COVID-19, the breakpoint for the ITS analysis of resistant isolates was set in March 2020.

Poisson (linear) segmented regression was used to analyze the data, with the statistical model being as follows:(1)$$ln\;P_{t}=\beta_{0}+\beta_{1}T_{t}+\beta_{2}X_{t}+\beta_{3}X_{t}T_{t}+\varepsilon_{t}$$
(2)$$R_{t}=\beta_{0}+\beta_{1}T_{t}+\beta_{2}X_{t}+\beta_{3}X_{t}T_{t}+\varepsilon_{t}$$
(3)$$\varepsilon_{t}=\rho \varepsilon_{t-1}+\mu_{t}$$

$P_{t}$ and $R_{t}$ are the outcome indicators of this model, representing the number of pathogenic bacteria detected and the antimicrobial resistance rate of isolates at each interval point *t*, respectively. $T_{t}$ is the time variable (in a fortnight), with 13 April 2018–26 April 2018 = 1, 27 April 2018–10 May 2018 = 2, and 11 May 2018–24 May 2018 = 3. $X_{t}$ is the binary variable used to distinguish the period before and after COVID-19, and $X_{t}T_{t}$ is the interaction term. $\beta_{0}$ represents the baseline level; $\beta_{1}$ indicates the slope or trend of pathogen detection level before COVID-19; $\beta_{2}$ and $\beta_{3}$ represent the change in the level and trend of the outcome indicators (compared with $\beta_{0}$ and $\beta_{1}$) since the start of COVID-19, indicating the short-term and long-term effects of COVID-19, respectively. $\varepsilon_{t}$ is a residual error which satisfies Equation (3) when following the first-order auto-correlation process. $\left|\rho \right|<1;\;\mu_{t}$ is normally distributed. The significance of the regression coefficients was tested using *z*-tests or *t*-tests, and the Durbin–Watson (DW) method was used to test for first-order autocorrelation, with DW values close to two indicating no autocorrelation. We used the Prais–Winsten algorithm to adapt the model to account for autocorrelation. Furthermore, in order to verify the results’ reliability applying duplicate isolates per patient during the analysis, we re-established all the models by keeping only the first isolate of the same strain per patient in a unit time. We conducted a stratified analysis to calculate the odds ratios of carbapenem resistance rates of CRKP and CRAB before and after the COVID-19 outbreak in different subgroups using the Cochran–Mantel–Haenszel (CMH) test.

R version 4.1.2 (Vienna, Austria) was used to build segmented Poisson and linear regression models for the interrupted time series analysis. IBM SPSS version 26.0 (Armonk, NY, USA) was used for the Chi-square test and CMH test. A *p*-value of <0.05 (two-tailed) was considered statistically significant.

## 5. Conclusions

This interrupted time series study was designed to examine the epidemiological characteristics of CROs and changes in antimicrobial resistance before and after COVID-19, providing hospitals with evidence for implementing targeted prevention and control measures against hospital-acquired infections during the pandemic. The distribution of CROs in teaching hospitals varies at the population, department, and ward levels. The resistance rate of CRAB increased after the COVID-19 outbreak, with the long-term trend showing a decrease in the CRAB resistance rate. However, the overall resistance of *K. pneumoniae* did not change significantly during COVID-19. Further stratified analysis revealed that the resistance rate of CRKP increased in the >65-year-old group, sputum sample group, and neurology group, which should be taken into consideration when developing hospital policies on antibiotics or infection prevention and control.

## Figures and Tables

**Figure 1 antibiotics-12-00431-f001:**
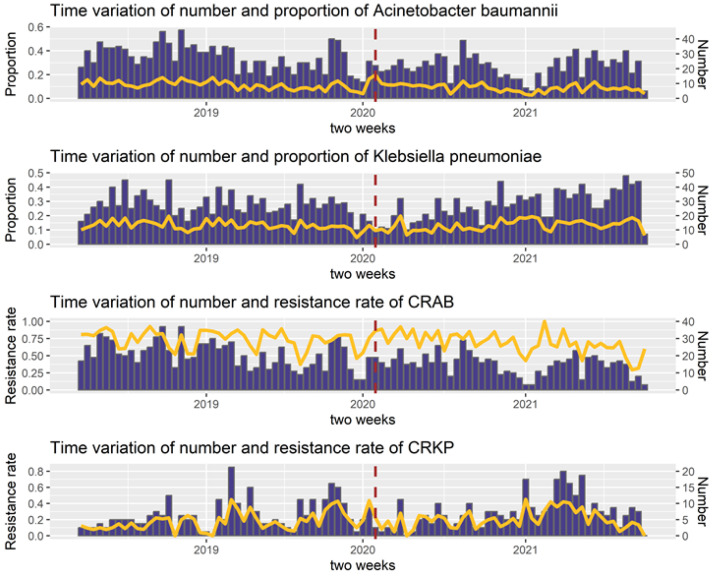
Time trends of pathogens and drug-resistant bacteria. Data were collected on a biweekly basis. The dark red dashed line indicates the COVID-19 outbreak. The yellow line represents the proportion of the pathogen or resistance rate of the drug-resistant bacteria. The blue bars show the number of pathogenic or drug-resistant bacteria.

**Figure 2 antibiotics-12-00431-f002:**
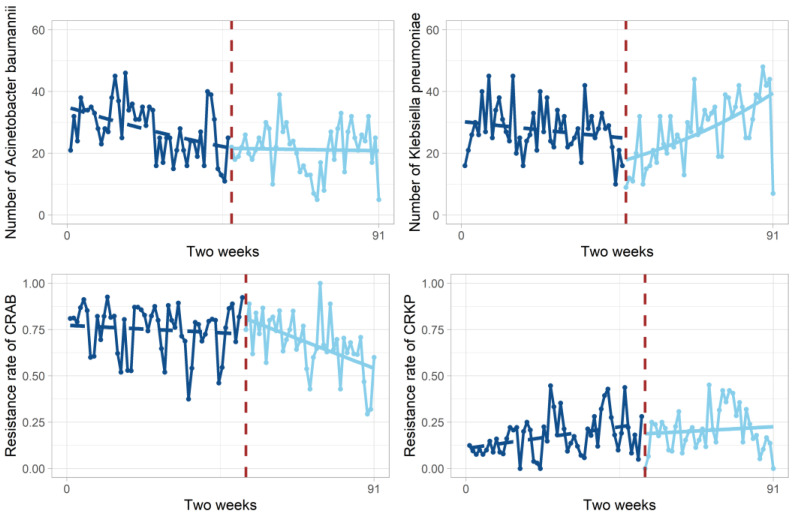
The change in outcome indicators every two weeks before and after COVID-19 outbreak. The dark blue dots represent the number (or resistance rate) before COVID-19 outbreak. The blue dots represent the number (or resistance rate) after COVID-19 outbreak. The dark blue dashed line represents the regression line before COVID-19 outbreak. The blue line represents the regression line after COVID-19 outbreak. Dark red dashed lines indicate breakpoints in the time series.

**Figure 3 antibiotics-12-00431-f003:**
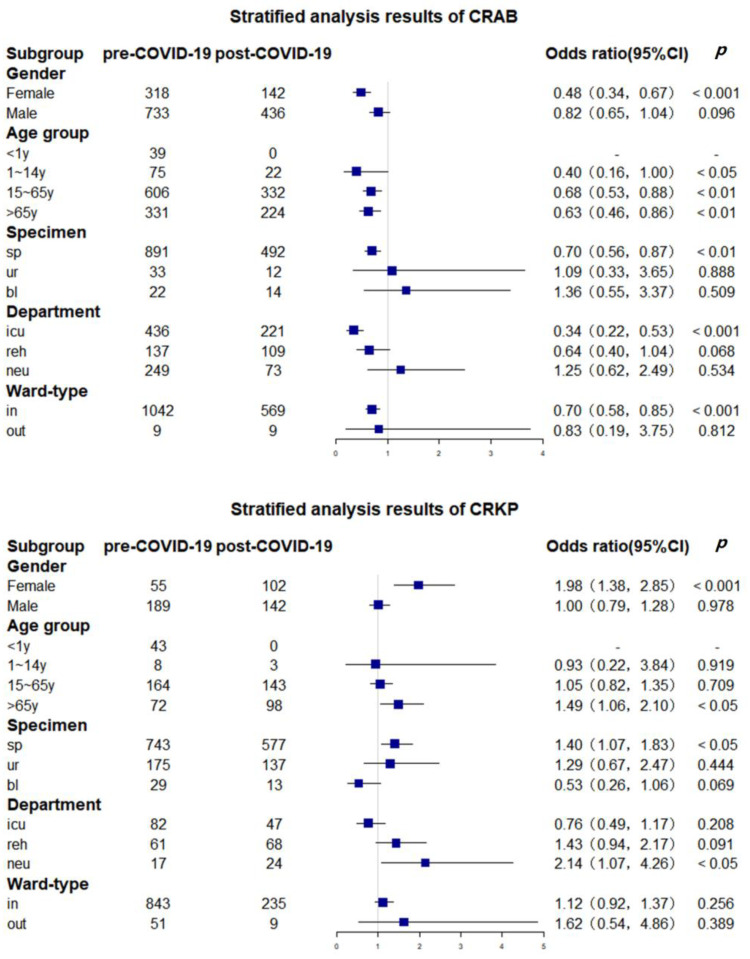
The change in CROs before and after COVID-19 in different subgroups. The sample size of outpatients was too small to be included in the stratified analysis. Abbreviations: sp, sputum; ur, urine; bl, blood; icu, intensive care unit; reh, rehabilitation center; neu, neurology.

**Table 1 antibiotics-12-00431-t001:** Proportions of common pathogenic bacteria in the teaching hospital.

	2018		2019		2020		2021		Total
	n (% ^a^)	CARSS	n (%)	CARSS	n (%)	CARSS	n (%)	CARSS	
* **K. pneumoniae** *	535 (11.78%)	465,322 (14.39%)	722 (11.01%)	503,230 (14.26%)	572 (9.94%)	482,330 (14.84%)	666 (12.77%)	-	2495 (11.31%)
* **A. baumannii** *	600 (13.21%)	227,091 (7.02%)	667 (10.17%)	239,890 (6.80%)	557 (9.70%)	219,921 (6.77%)	416 (7.97%)	-	2240 (10.15%)
**Number of isolates of all species**	4542	3,234,372	6558	3,528,471	5745	3,249,123	5217	-	22062

Note. ^a^ The number of *Klebsiella pneumoniae* (*Acinetobacter baumannii*) detected in a given year/the total number of isolates of all species in the hospital in that year. Abbreviations: *K. pneumoniae*, *Klebsiella pneumoniae*; *A. baumannii*, *Acinetobacter baumannii*; CARSS, China Antimicrobial Resistance Surveillance System.

**Table 2 antibiotics-12-00431-t002:** Results of the segmented regression analysis of number of *A. baumannii* and *K. pneumoniae* and the resistance rates of CRAB and CRKP.

Outcomes	Coefficient	Standard Error	*t* (*z*)	*p*-Value
Number of *Acinetobacter baumannii* detected (DW = 1.357)				
Baseline level (*β*_0_)	3.5561	0.0532	66.822	<0.001
Baseline trend (*β*_1_)	−0.0103	0.0021	−4.891	<0.001
Level change after COVID-19 (*β*_2_)	**−0.4436**	**0.1894**	**−2.343**	<0.05
Trend change after COVID-19 (*β*_3_)	**0.009**	**0.0033**	**2.796**	<0.01
Resistance rate of CRAB (DW = 1.949)				
Baseline level (*β*_0_)	0.7712	0.0443	17.404	<0.001
Baseline trend (*β*_1_)	−0.0008	0.0015	−0.527	0.600
Level change after COVID-19 (*β*_2_)	**0.3873**	**0.1685**	**2.299**	<0.05
Trend change after COVID-19 (*β*_3_)	**−0.0060**	**0.0027**	**−2.234**	<0.05
Number of *Klebsiella pneumoniae* detected (DW = 1.826)				
Baseline level (*β*_0_)	3.4125	0.0552	61.831	<0.001
Baseline trend (*β*_1_)	−0.0041	0.0021	−2.006	<0.05
Level change after COVID-19 (*β*_2_)	**−1.4185**	**0.1786**	**−7.944**	**<0.001**
Trend change after COVID-19 (*β*_3_)	**0.0226**	**0.0031**	**7.318**	**<0.001**
Resistance rate of CRKP (DW = 1.993)				
Baseline level (*β*_0_)	0.1031	0.0432	2.383	<0.05
Baseline trend (*β*_1_)	0.0029	0.0014	2.060	<0.05
Level change after COVID-19 (*β*_2_)	0.0115	0.1630	0.071	0.944
Trend change after COVID-19 (*β*_3_)	−0.0017	0.0026	−0.657	0.513

Note. DW, Durbin–Watson coefficient. $\beta_{2}$ and $\beta_{3}$ represent the change in the level and trend of the outcome indicators (compared with $\beta_{0}$ and $\beta_{1}$) since the start of COVID-19. *p* value of pathogenic bacteria was calculated by Z test. *p* value of isolates resistance was calculated by *t*-test. *p* < 0.05 is considered statistically significant. Bold text represents significant results after the COVID-19 outbreak.

## Data Availability

The data presented in this study are available on request from the corresponding author.

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
