# Peer review of "Epidemiological Characteristics and Antimicrobial Resistance Changes of Carbapenem-Resistant Klebsiella pneumoniae and Acinetobacter baumannii under the COVID-19 Outbreak: An Interrupted Time Series Analysis in a Large Teaching Hospital"

_antibiotics, 2023, doi:10.3390/antibiotics12030431_

Round 1

Reviewer 1 Report

In this study, the authors examined changes in the detection rate of Carbapenem-resistant organisms (CROs) before and after the COVID-19 outbreak. The detection rate of Carbapenem-resistant Acinetobacter baumannii (CRAB) was 38.73% higher in post-COVID-19 period than before. Furthermore, Carbapenem-resistant Klebsiella pneumoniae (CRKP) rate increased in females, > 65-year-old group, sputum sample group and neurology group. These results suggest that changes in CROs need to be carefully monitored in the high-risk populations and clinical departments. 

Overall, the manuscript is technically sound and the research ideas appear justified. The work is technically sound and is well organized. Nevertheless, the following are several comments regarding the submitted manuscript:

1. Figure 1 should describe what the yellow lines and dark blue bars indicate.

2. When you say, “resistance evolution”, do you mean “the mechanism by which resistance is acquired”? If this is the case, you should discuss the mechanism by which CROs acquire resistance.

3. Information on the use of antibacterial drugs is critical in the discussion of the effect of the COVID-19 pandemic on nosocomial infections and resistance in CROs. Is there any information on the use of antibacterial drugs at all?

Author Response

Response to Reviewer 1 Comments

In this study, the authors examined changes in the detection rate of Carbapenem-resistant organisms (CROs) before and after the COVID-19 outbreak. The detection rate of Carbapenem-resistant Acinetobacter baumannii (CRAB) was 38.73% higher in post-COVID-19 period than before. Furthermore, Carbapenem-resistant Klebsiella pneumoniae (CRKP) rate increased in females, > 65-year-old group, sputum sample group and neurology group. These results suggest that changes in CROs need to be carefully monitored in the high-risk populations and clinical departments.

Overall, the manuscript is technically sound and the research ideas appear justified. The work is technically sound and is well organized. Nevertheless, the following are several comments regarding the submitted manuscript:

Response: We appreciate the encouraging comments.

1. Figure 1 should describe what the yellow lines and dark blue bars indicate.

Response: Thanks for the advice. The descriptions have been added to the notes of Figure 1.

2. When you say, “resistance evolution”, do you mean “the mechanism by which resistance is acquired”? If this is the case, you should discuss the mechanism by which CROs acquire resistance.

Response: Thanks for the advice. Sorry for the ambiguity caused by the statement “resistance evolution”; we have corrected it to “resistance change”. In fact, we focus on antimicrobial resistance rate changes before and after COVID-19 outbreak rather than evolution mechanism.

3. Information on the use of antibacterial drugs is critical in the discussion of the effect of the COVID-19 pandemic on nosocomial infections and resistance in CROs. Is there any information on the use of antibacterial drugs at all?

Response: We agree. Information on antimicrobial agents is unavailable for us due to data limitations. This issue has been stressed as a limitation in the Discussion part, and we discussed the potential effects of antimicrobial agents. As you suggested, adding antibacterial drugs data can better explain the reasons for the changes in resistance of CROs, so further research should be carried out in the future when drug data is available.

Reviewer 2 Report

Although the study design is intriguing and stimulates epidemiologically useful observations, in our opinion there are methodological shortcomings.

The authors in the title refer to Carbapenem-resistant organisms, but in the text they describe only klebsiella pneumoniae and Acinetobacter baumannii, neglecting other gram negative bacteria such as Pseudomonas and Enterobacteriacae.

The authors generically define Carbapenem-resistant organisms, but the type of class that defines resistance (Ambler class A, B, C or D) has not been defined and analyzed. A wide range of enzymes have been identified among carbapenemase-producing Enterobactererales (KPC, MBL, NDM, VIM....). The authors should, therefore, include all carbapenem-resistant bacteria and specify which classes are present.

The authors should clarify how the isolates were identified. By which microbiological techniques? Likewise the authors should specify how the antibiotic sensitivity tests were performed.

The statistical analysis will need to be reconsidered in the light of updated data

Author Response

Response to Reviewer 2 Comments

Although the study design is intriguing and stimulates epidemiologically useful observations, in our opinion there are methodological shortcomings.

Response: We appreciate the critical comments and valuable suggestions. We have learned carefully and have made significant revisions in this revised manuscript.

The authors in the title refer to Carbapenem-resistant organisms, but in the text they describe only klebsiella pneumoniae and Acinetobacter baumannii, neglecting other gram negative bacteria such as Pseudomonas and Enterobacteriacae.

Response: Thanks for the advice. Considering that the title may be misleading, we decided to replace “carbapenem-resistant organisms” with “Klebsiella pneumoniae and Acinetobacter baumannii”. We chose these two bacteria because Klebsiella pneumoniae and Acinetobacter baumannii are typical carbapenem-resistant Enterobacteriaceae (CRE) and glucose non-fermenting (NF) carbapenem-resistant organisms, respectively. Furthermore, these two drug-resistant bacteria are also one of the "critical priority" drug-resistant bacteria identified by WHO, which cause heavy disease burden. Therefore, we chose to focus on these two species instead of all CROs.

Title: Epidemiological characteristics and antimicrobial resistance change of carbapenem-resistant Klebsiella pneumoniae and Acinetobacter baumannii under the COVID-19 outbreak: An interrupted time series analysis in a large teaching hospital

The authors generically define Carbapenem-resistant organisms, but the type of class that defines resistance (Ambler class A, B, C or D) has not been defined and analyzed. A wide range of enzymes have been identified among carbapenemase-producing Enterobactererales (KPC, MBL, NDM, VIM....). The authors should, therefore, include all carbapenem-resistant bacteria and specify which classes are present.

Response: We agree. The information on the type of class or enzymes that defines resistance was unavailable, which hindered us from performing more detailed analyses. In the dataset we collected, it is unable to tell different enzymes among the carbapenemase-producing Enterobactererales. We agree that it is valuable to present different classes of defines resistance. We have declared this issue as a limitation in the Discussion section.

The authors should clarify how the isolates were identified. By which microbiological techniques? Likewise the authors should specify how the antibiotic sensitivity tests were performed.

The statistical analysis will need to be reconsidered in the light of updated data.

Response: Thanks for the advice. The VITEK MS bacterial mass spectrometry was used for isolate identification and the Vitek 2 Compact system was used for antimicrobial susceptibility testing. Vitek MICs was classified according to Clinical and Laboratory Standards Institute guidelines. We have added the information to the section 4.2.

This study focuses on Acinetobacter baumannii and Klebsiella pneumoniae, two typical CROs, and type of class or enzymes that defines resistance was unavailable; therefore, it could not update the statistical analysis.

Reviewer 3 Report

Congratulations, Nice job!

Author Response

Congratulations, Nice job!

Response: We very appreciate the encouraging comment. Thanks!

Reviewer 4 Report

The paper addresses an important and currently very interesting topic. The methodology with time series analysis is appropriate, but there is a major flaw in the study design otherwise:

-          It is not clear what kind of isolates were included in the study: surveillance culture or diagnostic isolates? How many isolates per patient were included in the study, just first isolates or all isolates?

-          The number of isolates is not normalized, the authors only report the absolute numbers, the increase or decrease in patient population, that is in general typical for the pandemic (less elective surgery…)

If the authors do not improve this part of the study, the paper cannot be published.

In addition: the paper should be shortened and focus to the changes related to the pandemic.

Some more detailed comments below

Abstract

line 24: what kind of isolates were collected? Not clear but important to be mentioned in the abstract

Results section: what is detection rate? What are the absolute numbers, not %?

Line 31: for CrAb the authors write about detection rate, then they change to resistance rate in Klebsiella: the text should be more concise

Antibiotics' names should not be written with capital letters

One of the reasons for the increase of antimicrobial resistance is also lower adherence to infection control in crowded wards and shortage of staffing – the authors should address this aspect briefly

Materials and methods:

The comments in the Setting section are redoundant (the two last sentences, the last sentence is also not clear).

Data collection: the second sentence is not clear and not concise. It si not clear if the authors included the isolates from surveillance samples or diagnostic specimens. How many isolates did they include per patient? Were only the first isolates included or they included repetitive samples in one patient? The data on the number of detected isolates should be normalized (per 1000 patients for example) to allow comparison. More patients /or less admitted could be related to more or less isolates and resistant isolates.

Statistical analysis: what do the authors mean by detection rates? Sentence 295-6 is redundant.

Results

Table 2 and 3: merging the same species together in one table would be more informative than presenting susceptible and resistant isolates. It would be interesting to see if the epidemiology of resistant strains differs from the epidemiology of susceptible ones or total.

In general, the data presented in table 2 and 3 does not seem to be very relevant for the general idea of the paper – I suggest that they are omitted, that would make the paper more concise

Discussion

As in the Results section, the authors should focus on the impact of COVID-19 and not present/discuss general epidemiology of isolates.

The sentence starting in line 211: could the authors provide references for their statement? Otherwise, this sentence is redundant

Author Response

Response to Reviewer 4 Comments

The paper addresses an important and currently very interesting topic. The methodology with time series analysis is appropriate, but there is a major flaw in the study design otherwise:

Response: Many thanks. We appreciate the critical comments and valuable suggestions. We have learned carefully and have made targeted modifications in this revised manuscript.

- It is not clear what kind of isolates were included in the study: surveillance culture or diagnostic isolates? How many isolates per patient were included in the study, just first isolates or all isolates?

Response: Thanks for your comments. The type of isolates was diagnostic isolates from the patients and we included all isolates per patient in the database. Totally, there are 8697 patients with detected isolates from April 2018 to September 2021. Patients with only one or two isolates account for 80%, and patients with less than 5 isolates account for 94%. During the study period, patients may have more than one outpatient or inpatient visit.

- The number of isolates is not normalized, the authors only report the absolute numbers, the increase or decrease in patient population, that is in general typical for the pandemic (less elective surgery…).

If the authors do not improve this part of the study, the paper cannot be published.

Response: Thanks for your advice. We agree. Due to the unavailability of the number of patients visited the hospital during the study period, the number of detected isolates could not be normalized to per 1000 patients. Therefore, in this study, we intend to compare the resistance rate (%) of CRAB and CRKP rather than the exact number of isolates. Resistance rate is more appropriate to make comparisons, so we draw the main findings on this aspect and revised the relevant contents to avoid the misleading information. This is a limitation in our study, and we also clarified this issue in the limitation section.

In addition: the paper should be shortened and focus to the changes related to the pandemic.

Response: Thanks for the advice. We have revised the article, especially the results and discussion section, to focus more on the interpretation of changes related to COVID-19.

Some more detailed comments below

Abstract

line 24: what kind of isolates were collected? Not clear but important to be mentioned in the abstract.

Response: Thanks. We have clarified in the abstract. The antibiotic susceptibility tests in this study included all diagnostic isolates among the patients.

Results section: what is detection rate? What are the absolute numbers, not %?

Response: Thanks for your question. For pathogens, the detection rate refers to the percentage of pathogenic bacteria in different groups, which was the number of detected specific pathogenic bacteria in specific subgroup/the number of detected all pathogenic bacteria in the same subgroup ×100%, used to reflect the proportion of bacteria in a given subgroup. For resistant bacteria, “detection rate” refers to the resistance rate, and in case of misunderstanding, we have unified all the expressions as resistance rate. Because the total number of different groups was not the same, absolute numbers of isolates were not used for comparison. Both absolute numbers and resistance rates are presented in the table 2 and table 3.

Line 31: for CrAb the authors write about detection rate, then they change to resistance rate in Klebsiella: the text should be more concise

Response: Thanks. We have revised the manuscript to keep it consistent and concise.

Antibiotics' names should not be written with capital letters

Response: Sorry for the writing error. Corrected as per advice.

One of the reasons for the increase of antimicrobial resistance is also lower adherence to infection control in crowded wards and shortage of staffing – the authors should address this aspect briefly

Response: Thanks. We agree. We have addressed it briefly in the Abstract and expanded this point in the discussion section.

Materials and methods:

The comments in the Setting section are redoundant (the two last sentences, the last sentence is also not clear).

Response: Thanks for your advice. We deleted the penultimate sentence and revised the last sentence as suggested.

Data collection: the second sentence is not clear and not concise. It si not clear if the authors included the isolates from surveillance samples or diagnostic specimens. How many isolates did they include per patient? Were only the first isolates included or they included repetitive samples in one patient? The data on the number of detected isolates should be normalized (per 1000 patients for example) to allow comparison. More patients /or less admitted could be related to more or less isolates and resistant isolates.

Response: Thanks for your question. We agree. Please refer to our response to your comments 2 to 3 above. The isolates were from diagnostic specimens, and we included all diagnostic samples in one patient. In this study, we intend to compare the resistance rate of CRAB and CRKP rather than the exact number of isolates. Resistance rate (%) is more appropriate to make comparisons, so we draw the main findings on this aspect and revised the relevant contents to avoid the misleading information.

Statistical analysis: what do the authors mean by detection rates? Sentence 295-6 is redundant.

Response: Corrected as per advice. We have changed “detection rates” to “resistance rates” to avoid misunderstanding. The redundant sentences have been deleted.

Results

Table 2 and 3: merging the same species together in one table would be more informative than presenting susceptible and resistant isolates. It would be interesting to see if the epidemiology of resistant strains differs from the epidemiology of susceptible ones or total.

Response: Thanks for your advice. Corrected as per advice. We have put the combined results of Tables 2 and 3 in the supplementary material under the name “Appendix 1”.

In general, the data presented in table 2 and 3 does not seem to be very relevant for the general idea of the paper – I suggest that they are omitted, that would make the paper more concise

Response: Thanks for your advice. As suggested, we have removed the combined results of Tables 2 and 3 in the main text, and put it in the supplementary material.

Discussion

As in the Results section, the authors should focus on the impact of COVID-19 and not present/discuss general epidemiology of isolates.

Response: Thanks. Corrected as per advice. The general epidemiological characteristics have been truncated, and discussion is now more relevant to epidemiology impact of COVID-19 on isolates.

The sentence starting in line 211: could the authors provide references for their statement? Otherwise, this sentence is redundant

Response: Corrected as per advice. The sentence was not adequately supported by our results and has been removed.

Round 2

Reviewer 2 Report

The authors responded to any clarification made. Unfortunately, the lack of the required micrbiological data  reduces the clinical interst, altough the article is written in an appropriate way. The findings have more epidemiological then clinical significancew

Author Response

Response to Reviewer 2 Comments (Round 2)

Comments and Suggestions for Authors

The authors responded to any clarification made. Unfortunately, the lack of the required micrbiological data reduces the clinical interst, altough the article is written in an appropriate way. The findings have more epidemiological then clinical significancew

Response: Thanks. We very appreciate the critical comments and agree with the reviewer. This study mainly contributes to exploring the epidemiological characteristics and resistance change of carbapenem-resistant organisms (CROs) during the COVID-19 pandemic from an epidemiological perspective. As suggested, we would be able to conduct more clinically valuable analysis if detailed microbiological data were available.

Reviewer 4 Report

Dear authors,

you have tried to answer to most of my questions, but some of the most relevant (only one isolate per patient in a given time), and normalisation of the number of isolates remained unaswered, and the paper not improved accordingly. Because of the two problems - at least the issue of one isolate per patient in a given time could be solved - the quality of the paper remains rather low in spite of interesting topic and otherwise correct approach.

Table 1 should be further improved, add a line and a column that will help the reader understand what do the % refer to = what is 100%.

"resistance rates" (the % of resistant isolates?) and "detection rate" (n of isolates in two weeks?) should be used really consistently.

English language should be substantially improved, especially in the new parts of the text and in the abstract

Author Response

Response to Reviewer 4 Comments (Round 2)

Comments and Suggestions for Authors

Dear authors,

you have tried to answer to most of my questions, but some of the most relevant (only one isolate per patient in a given time), and normalisation of the number of isolates remained unaswered, and the paper not improved accordingly. Because of the two problems - at least the issue of one isolate per patient in a given time could be solved - the quality of the paper remains rather low in spite of interesting topic and otherwise correct approach.

Response: We very appreciate all your questions and comments, which help improve the manuscript quality. Now we have established all the models by keeping only the first isolate of the same strain per patient in a given time. Descriptive results were shown in Figure S1, and all model results were shown in the Table S2. The findings were consistent before and after the modification, although the coefficients of the models changed somewhat. The reduction of some sample sizes may have contributed to the statistical insignificance of only level change after COVID-19 for the number of Acinetobacter baumannii detected. Totally from April 2018 to September 2021, patients with only one or two isolates account for 80%, and patients with less than five isolates account for 94%, besides these patients have more than one outpatient or inpatient visit over several years. Since we set two weeks as the time unit for analysis, and detailed information of number of patients’ visits is unavailable, we are unable to make normalization of the number of isolates. We hope that some of the real-world data reflected in this hospital will give international readers some inspiration. Given the critical issue of antimicrobial resistance change during the COVID-19 pandemic, more research is needed.

Table 1 should be further improved, add a line and a column that will help the reader understand what do the % refer to = what is 100%.

Response: Many thanks for the advice. We added the summary row and column and annotated the meaning of % according to your suggestions.

"resistance rates" (the % of resistant isolates?) and "detection rate" (n of isolates in two weeks?) should be used really consistently.

Response: Thanks for the valuable comments. The “resistance rate” refers to the percentage of resistant isolates in the total number of specific isolates. The “detection rate” refers to the percentage of pathogenic bacteria in different groups within a specific period. In particular, to achieve the modeling needs, the unit time was set to two weeks, so the “detection rate” and “resistance rate” were calculated within two weeks. In the overall descriptive results, the time scales for detection and resistance rates were the entire study period. The relevant statements were checked carefully for consistency.

English language should be substantially improved, especially in the new parts of the text and in the abstract

Response: Thanks. We have checked carefully and improved the English language throughout the manuscript, especially the parts the reviewer suggested.

Round 3

Reviewer 4 Report

The paper is improved in comparison to the first version. There are still some inconsistencies in reporting detection rates instead of number of isolates in some parts of the text. Since the authors cannot provide the denominators (number of patients) we cannot use the word "rate".

English language still needs some improvement.

the discusion is too long.

Author Response

Response to Reviewer 4 Comments

The paper is improved in comparison to the first version. There are still some inconsistencies in reporting detection rates instead of number of isolates in some parts of the text. Since the authors cannot provide the denominators (number of patients) we cannot use the word "rate".

Response: Thanks for the valuable comments. We quite agree with this view. Given that the indicator in Table S1 does not provide the total number of patients and its numerator is part of the denominator, it is better to use “proportion” rather than “rate”. We have revised the expression in the text.

English language still needs some improvement.

Response: Many thanks. As suggested, we put many efforts on improving the English language. We have checked carefully, improved the article's wording, grammar, and fluency, and invite a native English-speaking collaborator to check.

the discusion is too long.

Response: Thanks. We have shortened the length of the discussion part. Given that the focus of the study was on changes in antimicrobial resistance under COVID-19, we appropriately reduced the discussion of general epidemiological characteristics.